# A Physical Measure for Characterizing Crossover from Integrable to Chaotic Quantum Systems

**DOI:** 10.3390/e25020366

**Published:** 2023-02-17

**Authors:** Chenguang Y. Lyu, Wen-Ge Wang

**Affiliations:** 1Department of Modern Physics, University of Science and Technology of China, Hefei 230026, China; 2CAS Key Laboratory of Microscale Magnetic Resonance, University of Science and Technology of China, Hefei 230026, China

**Keywords:** integrability-chaos cross-over, quantum chaos, eigenstate statistics, transition probability, Lipkin-Meshkov-Glick (LMG) model

## Abstract

In this paper, a quantity that describes a response of a system’s eigenstates to a very small perturbation of physical relevance is studied as a measure for characterizing crossover from integrable to chaotic quantum systems. It is computed from the distribution of very small, rescaled components of perturbed eigenfunctions on the unperturbed basis. Physically, it gives a relative measure to prohibition of level transitions induced by the perturbation. Making use of this measure, numerical simulations in the so-called Lipkin-Meshkov-Glick model show in a clear way that the whole integrability-chaos transition region is divided into three subregions: a nearly integrable regime, a nearly chaotic regime, and a crossover regime.

## 1. Introduction

In the past near half century, a large number of topics have been studied in the field of quantum chaos and a huge amount of knowledge about properties of quantum chaotic systems has been accumulated (see, e.g., [1,2]). However, this field is far from being fully explored. For example, although the spectral statistics of quantum chaotic systems have been studied well [3,4,5,6,7,8,9], not so much is known about wave functions, particularly about statistical properties of energy eigenfunctions (EFs) as expansions of energy eigenstates in certain bases.

As is well known, in classical mechanics, chaos refers to trajectory sensitivity to initial conditions, characterized by positive (maximum) Lyapunov exponents.The percentage of phase space, which is occupied by chaotic trajectories, supplies a useful quantitative measure in the study of crossover from integrability to chaos, at least for systems in a two-dimensional configuration space. However, the story is much more complicated with quantum systems and the route (from integrability) to quantum chaos is far from being fully understood.

Quantitative characterization of crossover from quantum integrability to quantum chaos is a topic that is of importance in many aspects. Indeed, without a deep understanding of mechanisms that may lead to a break down of integrability, it is unimaginable that a complete understanding of quantum chaos may be achieved; and, it is usually believed that quantum chaos may play a crucial role in establishing a sound foundation for quantum statistical mechanics and for understanding thermalization processes [10,11,12,13,14,15,16]. Many quantities have been studied for the characterization; loosely speaking, they may be classified into three classes: spectral statistics (see, e.g., Refs. [2,3,4,17,18,19,20,21]), statistical properties of EFs (see, e.g., Refs. [14,22,23,24,25,26,27,28,29,30,31,32,33]), and time evolution properties (see, e.g., Refs. [34,35,36,37,38,39,40,41,42,43,44,45]). Unfortunately, none of them is as satisfactory as the above-mentioned measure in the classical case.

Regarding spectral statistics, the nearest-level-spacing distribution P(s) is often studied. Its shape is close to the Poison distribution $e^{-s}$ in generic integrable systems [17], while, is close to the Wigner-Dyson distribution $P_{W}\left(s\right)=\frac{\pi }{2}s\exp\left(-\frac{\pi }{4}s^{2}\right)$ in quantum chaotic systems, the latter of which is almost identical to the prediction of random matrix theory (RMT) (for systems with the time-reversal symmetry) [2,3,4]. Among possible interpolations between Poison and Wigner-Dyson distributions [18,19,20,21], the most often studied one is the so-called Brody distribution [18], characterized by a Brody parameter β, with β=0 for integrable systems and β=1 for chaotic ones. Unfortunately, physical meaning of the Brody parameter is still unclear; in other words, it is regarded merely as a fitting parameter.

Properties of EFs may also be used for characterizing the crossover from quantum integrability to quantum chaos. To compute such properties, usually one needs to take a specific basis, which may be chosen for some physical reason or based on some mathematical consideration [14,25,26,27,28,29,30,31]. Recently, two methods were proposed [32,33], which employ intrinsic bases of the systems studied, namely, the eigenbases of their Hamiltonians. The first method makes use of the so-called adiabatic gauge potential (AGP) [32], describing the rate of deformation of eigenstates under infinitesimal perturbation. It was proposed that scaling behavior of the norm of AGP with system size may be used as an indicator for the integrability-chaos crossover. This indicator is basically qualitative and its validity is not completely clear.

The second method also studies the response of eigenstates to small perturbation, but, by making use of the distribution of rescaled components of perturbed states on the unperturbed basis [33]. This distribution has a Gaussian form in quantum chaotic systems, a phenomenon which may be traced back to the so-called Berry’s conjecture [22,23,24,30]; while, it deviates notably from the Gaussian form in quantum integrable systems. This method uses the difference between the distribution of rescaled components and the Gaussian distribution for the purpose of characterizing the integrability-chaos crossover. It suffers from two shortcomings: (i) Although the above-mentioned difference gives a quantitative measure to the distance to chaos, it is irrelevant to integrability; and, (ii) its physical meaning is not directly clear.

In this paper, we go further along the direction of the second method discussed above and make improvements. Specifically, we are to show that the value of the distribution of rescaled components at the origin point is a good candidate, i.e., it is of physical relevance and supplies a measure for characterizing the whole process from integrability to chaos.

The paper is organized as follows. A preliminary discussion is found in Section 2, about the class of systems to be studied, requirements of the type of perturbation to be employed, and some details of the two previous works mentioned above. The proposed measure for integrability-chaos crossover is discussed in Section 3 and its illustration in a model is given in Section 4. Finally, conclusions and discussions are given in Section 5.

## 2. Preliminary Discussions

In this section, we first discuss basic properties of the systems to be studied (Section 2.1), and then discuss properties of the perturbation to be considered (Section 2.2). Finally, we briefly recall basic contents of the two methods mentioned above, which use intrinsic bases for characterizing crossover from integrable to chaotic quantum systems (Section 2.3).

### 2.1. Perturbed and Unperturbed Systems

We study the response of Hamiltonian eigenstates to small perturbation. The unperturbed Hamiltonian is denoted by $H^{0}\left(\lambda \right)$, with a running parameter λ, such that it is integrable at λ=0 and is chaotic at λ in some region around 1. We use |k(λ)〉 to indicate eigenstates of $H^{0}\left(\lambda \right)$, with eigenenergies $E_{k}^{0}\left(\lambda \right)$ in the increasing energy order,
(1)$$H^{0}\left(\lambda \right)|k\left(\lambda \right)\rangle =E_{k}^{0}\left(\lambda \right)|k\left(\lambda \right)\rangle .$$Sometimes, when there is no risk of causing confusion, for brevity, the λ-dependence of $H^{0}$ and of its eigenstates is not written explicitly.

The perturbed Hamiltonian is written as
(2)$$H=H^{0}\left(\lambda \right)+\epsilon V,$$
where λ-dependence is not written explicitly, where ϵ is a very small parameter and *V* represents a perturbation operator. Eigenstates of *H* are denoted by |α〉,
(3)$$H|\alpha \rangle =E_{\alpha }|\alpha \rangle ,$$
with the eigenenergies $E_{\alpha }$ also in the increasing energy order. Components of the EF of a perturbed state |α〉 on the unperturbed basis are written as
(4)$$C_{\alpha k}=\langle k|\alpha \rangle .$$We assume that the Hilbert space is sufficiently large for meaningful statistical analysis of properties of EFs. For the simplicity in discussion, we consider only systems with the time-reversal symmetry, such that the components $C_{\alpha k}$ are real numbers. (It is straightforward to generalize results to be given below to the generic case without the time-reversal symmetry.)

We are to discuss properties of the distribution of rescaled $C_{\alpha k}$, denoted by ${\tilde{C}}_{\alpha k}$,
(5)$${\tilde{C}}_{\alpha k}:=\frac{C_{\alpha k}}{\sqrt{\langle |C_{\alpha k}{|}^{2}\rangle }},$$
where $\langle |C_{\alpha k}{|}^{2}\rangle$ indicates the average shape of EFs. We indicate this distribution by $g(\tilde{C})$. When computing $\langle |C_{\alpha k}{|}^{2}\rangle$, special attention should be paid to big components of the EFs. For example, for a chaotic system $H^{0}\left(\lambda \right)$, at a sufficiently small parameter ϵ, usually each perturbed state |α〉 has one big component of $C_{\alpha k}$; its value is close to 1, while other components of $C_{\alpha k}$ are much smaller, proportional to ϵ or smaller as predicted by the perturbation theory. In this case, clearly, if the biggest component is included, it usually makes $\langle |C_{\alpha k}{|}^{2}\rangle$ not smooth. Hence, in the computation of $\langle |C_{\alpha k}{|}^{2}\rangle$, the biggest components of $C_{\alpha k}$ should not be included and, consistently, they are not included in the distribution $g(\tilde{C})$, either.

Moreover, an average may be taken over those perturbed states that lie within a narrow energy shell, denoted by $\Gamma_{\alpha }$, which is centered at $E_{\alpha }$ and has a small width δe. To summarize, we write
(6)$$\langle |C_{\alpha k}{|}^{2}\rangle =\frac{1}{\sum_{E_{\beta }\in \Gamma_{\alpha }}^{\prime }1}{\sum_{E_{\beta }\in \Gamma_{\alpha }}}^{\prime }|C_{\beta k}{|}^{2},$$
where
(7)$$\Gamma_{\alpha }=\left[E_{\alpha }-\delta e/2,E_{\alpha }+\delta e/2\right]$$
and the prime over ∑ means that big components are excluded. (see Section 3.2 for further discussions on big components to be excluded in computations performed in integrable systems).

### 2.2. Properties of the Perturbation V

Physically, a prominent difference between a quantum integrable system and a quantum chaotic system lies in that the former has at least two independent good quantum numbers, while, the latter has only one which is related to the Hamiltonian. Due to this difference, at least to a certain type of perturbation, the response of an integrable system should be different from that of a chaotic system. Hence, in principle, response to perturbation may be used for the purpose of characterizing the integrability-chaos crossover. In this section, we discuss a type of perturbation *V* that may be used for this purpose.

It may be useful to provide a little discussion, in comparison with a purely mathematical viewpoint by which an operator *V* on an arbitrary basis is represented by a matrix with only one restriction — hermiticity. In fact, as is known, if no further restriction is imposed to the form of the matrix of *V*, there may exist some matrix of *V*, for which the components $C_{\alpha k}$ in an integrable system may show behaviors qualitatively similar to those in a chaotic system (see, e.g., Ref. [46]). Hence, for the purpose of “detecting” the difference between integrable and chaotic systems, a purely mathematical viewpoint is not enough. In other words, physical considerations should be taken into account, which may give certain restriction to the perturbation *V*.

To further analyze the above-discussed point, we discuss from the perspective of dynamic groups underlying models studied. Here, a group is called a dynamic group underlying a model, if the model Hamiltonian is a function of generators of the group. (A dynamic group is not necessarily a symmetry group. In other words, it is unnecessary for the model Hamiltonian to possess any symmetry related to the dynamic group.) We note that most physical models of realistic interest have some underlying dynamic Lie groups. Usually, generators of the dynamic group may take the form of raising and lowering operators, which we indicate as $K_{\eta }^{†}$ and $K_{\eta }$, respectively, with a label η. For example, for an oscillator in a one-dimensional configuration space, the so-called Weyl-Heisenberg group is the underlying dynamic group, where generators as raising and lowering operators are given by well-known combinations of the position *x* and momentum *p*.

On the integrable side of $H^{0}\left(\lambda \right)$ with λ=0, two different cases should be treated differently: interacting and noninteracting integrable systems. Let us first discuss a noninteracting integrable system, whose Hamiltonian $H^{0}\left(0\right)$ depends separately on generators of different degrees of freedom. Since there is no interplay between different degrees of freedom, usually the integrable states |k(0)〉 may be obtained by multiplying raising operators on certain “vacuum state” denoted by |0〉, i.e.,
(8)$$|k\left(0\right)\rangle =\prod_{\eta }{\left(K_{\eta }^{†}\right)}^{m_{\eta }}|0\rangle ,$$
where $m_{\eta }$ are nonnegative integers. Usually, the integers $m_{\eta }$ are, or are related to, good quantum numbers in the state |k(0)〉. On the basis of {|k(0)〉}, elements of *V* are written as
(9)$$\langle k^{\prime }\left(0\right)|V|k\left(0\right)\rangle =\prod_{\eta \eta^{\prime }}\langle 0|{\left(K_{\eta^{\prime }}\right)}^{m_{\eta^{\prime }}^{\prime }}V{\left(K_{\eta }^{†}\right)}^{m_{\eta }}|0\rangle .$$

One useful observation is that, for many types of perturbation *V* of physical interest, the matrix of *V* with elements given in Equation (Equation 9) has an interesting feature; i.e., it is sparse in the sense of possessing many zero elements of $\langle k^{\prime }\left(0\right)|V|k\left(0\right)\rangle$. As one example, one may consider perturbations that are controllable and describable in laboratories. Such a perturbation *V* is usually represented by some simple function of $\left\{K_{\eta }^{†},K_{\eta }\right\}$ (at least not a complicated function) and, as a result, Equation (Equation 9) predicts a sparse matrix of *V*. As another example, one may consider a perturbation *V*, which may bring a limited change to the good quantum numbers $m_{\eta }$ for most of the states |k(0)〉. In this case, one also finds a sparse structure of the matrix of $\langle k^{\prime }\left(0\right)|V|k\left(0\right)\rangle$.

Then, we discuss an interacting integrable system, for which generators of different degrees of freedom interplay in the Hamiltonian $H^{0}\left(0\right)$. In such system, the states |k(0)〉 are superpositions of product terms as given on the right-hand side of Equation (Equation 8). In this case, the simplicity of the function $V(K_{\eta }^{†},K_{\eta })$ does not guarantee a sparse structure of the matrix of $\langle k^{\prime }\left(0\right)|V|k\left(0\right)\rangle$. While, for a perturbation *V*, which may bring a limited change to good quantum numbers of the states |k(0)〉 (usually not the integers $m_{\eta }$), one still finds a sparse structure of the matrix of $\langle k^{\prime }\left(0\right)|V|k\left(0\right)\rangle$. (Mathematically, the simplest example of this type of operator *V* is written as $V=|k_{1}\left(0\right)\rangle \langle k_{2}\left(0\right)|+|k_{2}\left(0\right)\rangle \langle k_{1}\left(0\right)|$ with some fixed values of $k_{1}$ and $k_{2}$. Usually, to construct such an operator of physical interest is a model-dependent matter.)

On the chaotic side of $H^{0}\left(\lambda \right)$ with λ around 1, one finds a different story. To be specific in this discussion, let us consider a system that possesses a classical counterpart. To obtain an estimate, one may make use of Berry’s conjecture as a semiclassical prediction [22,23,24,30].

We first discuss $H^{0}\left(\lambda \right)$ of λ∼1, with respect to a noninteracting integrable system $H^{0}\left(0\right)$. Without loss of generality, we assume that the eigenstates |k(0)〉 in Equation (Equation 8) are also eigenstates of action. According to a version of Berry’s conjecture given on an action basis [30], the components 〈k(0)|k(λ)〉 have the following expression,
(10)$$\langle k\left(0\right)|k\left(\lambda \right)\rangle \propto \Omega_{k(0)k(\lambda )}R_{k(0)k(\lambda )},$$
where $R_{k(0)k(\lambda )}$ are Gaussian random variables (with mean zero and a normal distribution) and $\Omega_{k(0)k(\lambda )}$ indicates the so-called classical analog of the average shape of eigenfunctions, which is given by the overlap of the classical energy surface of $E_{k}^{0}\left(\lambda \right)$ and a classical torus corresponding to the action given by $m_{\eta }$. On the basis of |k(λ)〉, elements of *V* are written as
(11)$$\begin{array}{cc} \; & \langle k^{\prime }\left(\lambda \right)|V|k\left(\lambda \right)\rangle \\ & =\sum_{k\left(0\right),k^{\prime }\left(0\right)}\langle k^{\prime }\left(\lambda \right)|k^{\prime }\left(0\right)\rangle \langle k^{\prime }\left(0\right)|V|k\left(0\right)\rangle \langle k\left(0\right)|k\left(\lambda \right)\rangle \end{array}$$Making use of Equations (Equation 9) and (Equation 10), one finds that, at least for $E_{k^{\prime }}^{0}\left(\lambda \right)$ not far from $E_{k}^{0}\left(\lambda \right)$, the elements $\langle k^{\prime }\left(\lambda \right)|V|k\left(\lambda \right)\rangle$ are typically nonzero. Hence, the matrix of *V* on the chaotic basis {|k(λ)〉} has a structure, which is qualitatively different from the previously discussed sparse structure of *V* on the integrable basis {|k(0)〉}.

Next, we discuss $H^{0}\left(\lambda \right)$ of λ∼1, with respect to an interacting integrable system. In this case, the integrable states |k(0)〉 are superpositions of the action states. As a consequence, the components 〈k(0)|k(λ)〉 have an expression, which is more complicated than the right-hand side of Equation (Equation 10) for the noninteracting case. This implies that the elements $\langle k^{\prime }\left(\lambda \right)|V|k\left(\lambda \right)\rangle$ should be also typically nonzero at least for $E_{k^{\prime }}^{0}\left(\lambda \right)$ not far from $E_{k}^{0}\left(\lambda \right)$. Moreover, this is also true, even for a perturbation *V* that brings a limited change to good quantum numbers of the states |k(0)〉. Hence, as well, the matrix of *V* on the chaotic basis {|k(λ)〉} has a structure, which is qualitatively different from the previously discussed sparse structure of *V* on the integrable basis {|k(0)〉}.

To summarize, for a certain perturbation *V*, its matrix has a sparse structure on an integrable basis, while it does not have a similar structure on a chaotic basis. In Section 3, we are to focus on this type of perturbation *V* and use it to propose a quantity for characterizing integrability-chaos crossover.

### 2.3. Two Previously Studied Methods

In this section, we briefly recall basic contents of the two previously studied methods mentioned in the introduction, in which intrinsic bases are employed for characterizing integrability-chaos crossover.

The first method makes use of the AGP, which describes variation of eigenstates under infinitesimal change of the parameter λ. More exactly, denoted by $A_{\lambda }$, the AGP generates an adiabatic evolution of the eigenstates,
(12)$$A_{\lambda }|k\left(\lambda \right)\rangle =i\partial_{\lambda }|k\left(\lambda \right)\rangle .$$
Its offdiagonal elements satisfy
(13)$$\langle k|A_{\lambda }|l\rangle =-\frac{i}{E_{k}^{0}\left(\lambda \right)-E_{l}^{0}\left(\lambda \right)}\langle k|\partial_{\lambda }H^{0}\left(\lambda \right)|l\rangle$$
with k≠l; while, its diagonal elements may be set zero due to the freedom in choosing phases of the eigenstates, i.e., $\langle k|A_{\lambda }|k\rangle =0$ for all |k〉. (In Pandey et al. [32], to avoid a problem that may be caused by degeneracy of the spectrum of $H^{0}\left(\lambda \right)$, on the right-hand side of Equation (Equation 13), the term $\frac{1}{E_{k}^{0}-E_{l}^{0}}$ is replaced by $\frac{E_{k}^{0}-E_{l}^{0}}{{\left(E_{k}^{0}-E_{l}^{0}\right)}^{2}+\mu^{2}}$ with some small energy cutoff μ.) The Frobenius norm, also called Hilbert–Schmidt norm, of the AGP operator is written as (This norm is equal to the sum of the so-called fidelity susceptibility of the eigenstates |k(λ)〉 [47,48].)
(14)$$∥A_{\lambda }{∥}^{2}=\sum_{k,l}\left|\langle k\right|A_{\lambda }{\left|l\rangle \right|}^{2}.$$

It was proposed in Pandey et al. [32] that exponential scaling behavior of the AGP norm with respect to the particle number *N* of a many-body quantum system may be used as an indicator of quantum chaos, with polynomial behavior expected for integrable systems. For systems satisfying the so-called eigenstate thermalization hypothesis (ETH) (see Equation (Equation 24) to be given below), it is not difficult to check that the AGP norm indeed scales exponentially with the system size. However, the reverse statement is not supported by any analytical analysis and this makes the above proposal questionable as a criterion for quantum chaos. In fact, according to numerical simulations given in Pandey et al. [32] in an XXZ chain, the integrability-chaos crossover region predicted by the above-discussed proposal is much lower than that obtained by an ordinary method of employing spectral statistics, by at least one order of magnitude.

In the second method, the difference between the distribution $g(\tilde{C})$ and the Gaussian distribution is used as a measure for the distance to quantum chaos [33]. Quantitatively, the difference is written as $\Delta_{\mathrm{EF}}=\int \left|I\left(\tilde{C}\right)-I_{G}\left(\tilde{C}\right)\right|d\tilde{C}$, where $I(\tilde{C})$ is the cumulative distribution of $g(\tilde{C})$ and $I_{G}$ is the cumulative distribution for the Gaussian distribution of
(15)$$g_{G}\left(\tilde{C}\right)=\frac{1}{\sqrt{2\pi }}\exp\left(-{\tilde{C}}^{2}/2\right).$$

This difference is expected to be small in quantum chaotic systems, and large in integrable systems (see the next section for detailed discussions).

Numerical simulations performed in a three-orbital Lipkin-Meshkov-Glick (LMG) model [49] show that the quantity $\Delta_{\mathrm{EF}}$ behaves consistently with $\Delta_{W}$ in the regime of integrability-chaos crossover [33]. Here, $\Delta_{W}$ indicates the difference between the nearest-level-spacing distribution P(s) and the Wigner distribution $P_{W}\left(s\right)$, as a measure for the distance to chaos; more precisely,
(16)$$\Delta_{W}=\int \left|I\left(s\right)-I_{W}\left(s\right)\right|ds,$$
where “*I*” also indicate the corresponding cumulative distributions. It was found that integrability is manifested by large values of $\Delta_{\mathrm{EF}}$, which are due to high peaks of $g(\tilde{C})$ at $\tilde{C}=0$.

Finally, let us give a brief comparison of the two methods discussed above. The physics lying behind them are in fact related. This point is seen quite clearly, in the special case that $H^{0}\left(\lambda \right)=H^{0}\left(0\right)+\lambda V$. In this case, $\partial_{\lambda }H^{0}\left(\lambda \right)=V$ and the AGP elements in Equation (Equation 13) are directly related to the components $C_{\alpha k}$ divided by ϵ, which are predicted by a first-order perturbation theory [see Equation (Equation 19) to be given below].

Meanwhile, the two methods have important differences. First, the second method in fact employs the rescaled components ${\tilde{C}}_{\alpha k}$ in Equation (Equation 5). Without the rescaling procedure, the distribution of components would not be close to a Gaussian form in quantum chaotic systems. Second, it is not the AGP norm itself that is employed as a measure for integrability-chaos crossover, but it is the AGP scaling behavior with respect to system size. This implies that the first method does not supply a measure for characterizing the whole crossover region, though it might be used for detecting some “crossover point”, if in existence.

## 3. g(0) as a Crossover Measure

In this section, going further along the direction of the second method discussed above, we propose that $g(\tilde{C}=0)$ may be used as a measure for characterizing integrability-chaos crossover. Specifically, we discuss its physical meaning in Section 3.1 and discuss its detailed properties in Section 3.2.

### 3.1. A Physical Meaning of g(0)

We first consider a generic case, in which the unperturbed Hamiltonian $H^{0}\left(\lambda \right)$ possesses a nondegenerate spectrum. In this case, under a sufficiently small ϵ, the perturbed states |α〉 may be approximated by their first-order perturbation expansions. For a given state |α〉, we use $k_{\alpha }$ to indicate the label *k* for which $E_{k}^{0}$ is the closest to $E_{\alpha }$. Clearly, $C_{\alpha k_{\alpha }}≃1$. Then, one writes
(17)$$\left|\alpha \rangle ≃\right|\alpha^{\left(0\right)}\rangle +|\alpha^{\left(1\right)}\rangle ,$$
where $|\alpha^{\left(0\right)}\rangle =|k_{\alpha }\rangle$ and
(18)$$\begin{array}{cc} |\alpha^{\left(1\right)}\rangle = & \sum_{k\neq k_{\alpha }}\frac{\epsilon V_{kk_{\alpha }}}{E_{k_{\alpha }}^{0}-E_{k}^{0}}|k\rangle . \end{array}$$From Equation (Equation 18), one sees that
(19)$$C_{\alpha k}≃\frac{\epsilon V_{kk_{\alpha }}}{E_{k_{\alpha }}^{0}-E_{k}^{0}}\;\mathrm{for}\;k\neq k_{\alpha }.$$

To see a physical meaning of g(0), let us discuss the probability of transition from an arbitrary state $|k_{0}\rangle$ (as an initial state |ψ(t=0)〉) to an arbitrary state |k〉 ($k\neq k_{0}$) under a perturbation ϵV, which is given by ${\left|\langle k|\psi \left(t\right)\rangle \right|}^{2}$. It is easy to find that
(20)$$\langle k|\psi \left(t\right)\rangle =\langle k|e^{-iHt}|k_{0}\rangle =\sum_{\alpha }C_{\alpha k_{0}}C_{\alpha k}e^{-iE_{\alpha }t}.$$Since $C_{\alpha k}≃1$ for $k=k_{\alpha }$, one further obtains that
(21)$$\langle k|\psi \left(t\right)\rangle ≃C_{\alpha_{1}k_{0}}e^{-iE_{\alpha_{1}}t}+C_{\alpha_{2}k}e^{-iE_{\alpha_{2}}t},$$
where $\alpha_{1}$ is determined by the relation of $k_{\alpha_{1}}=k$ and $\alpha_{2}$ by the relation of $k_{\alpha_{2}}=k_{0}$. According to Equation (Equation 21), the smallness of the components $C_{\alpha_{1}k_{0}}$ and $C_{\alpha_{2}k}$, if compared with other unperturbed states, has a clear physical meaning; that is, it implies relative prohibition of the quantum transition from $|k_{0}\rangle$ to |k〉. However, a problem is met with quantitative characterization for this, because all the components $C_{\alpha k}$ of $k\neq k_{\alpha }$ go to zero in the limit of ϵ→0.

To circumvent the above-discussed problem, instead, we consider the rescaled components ${\tilde{C}}_{\alpha k}$ in Equation (Equation 5), which remain finite in the limit of ϵ→0. Making use of perturbation theory, it is not difficult to find that the two components $C_{\alpha_{1}k_{0}}$ and $C_{\alpha_{2}k}$, which are related to the same unperturbed energy difference of $\left(E_{k}^{0}-E_{k_{0}}^{0}\right)$, should have similar absolute values on average, i.e., $\langle |C_{\alpha_{1}k_{0}}{|}^{2}\rangle ≃\langle |C_{\alpha_{2}k}{|}^{2}\rangle$. Then, Equation (Equation 21) is written as
(22)$$\frac{\langle k|\psi (t)\rangle }{\sqrt{\langle |C_{\alpha_{1}k_{0}}{|}^{2}\rangle }}≃{\tilde{C}}_{\alpha_{1}k_{0}}e^{-iE_{\alpha_{1}}t}+{\tilde{C}}_{\alpha_{2}k}e^{-iE_{\alpha_{2}}t}.$$Thus, relative smallness of the rescaled components ${\tilde{C}}_{\alpha_{1}k_{0}}$ and ${\tilde{C}}_{\alpha_{2}k}$ implies relative smallness of the transition probability divided by the average shape of EFs.

From the above discussions, one sees that high population of small rescaled components usually implies strong prohibition of quantum transition. Quantitatively, the population is characterized by the value of the distribution $g(\tilde{C})$ at $\tilde{C}=0$, namely, by g(0).

Next, we discuss the special case, in which $H^{0}\left(\lambda \right)$ possesses a degenerate spectrum. In this case, for a given unperturbed state |k〉, the components $C_{\alpha k}$ of those perturbed states |α〉, whose energies $E_{\alpha }$ are close to $E_{k}^{0}$, may be large. We use $S_{k}$ to denote the set of the indices α of these perturbed states. For a nondegenerate level $E_{k}^{0}$, the set $S_{k}$ in fact contains one label α only, such as in the case of nondegenerate spectrum discussed above; while, for a degenerate level, the number of elements of $S_{k}$ is usually equal to the degeneracy. In most cases, *k* and $k_{0}$ do not belong to a same degenerate level (Study of the case, in which *k* and $k_{0}$ belong to a same degenerate level, is hard and is beyond the scope of perturbative analysis.) and Equation (Equation 20) gives that
(23)$$\langle k|\psi \left(t\right)\rangle ≃\sum_{\alpha_{1}\in S_{k}}C_{\alpha_{1}k_{0}}e^{-iE_{\alpha_{1}}t}+\sum_{\alpha_{2}\in S_{k_{0}}}C_{\alpha_{2}k}e^{-iE_{\alpha_{2}}t}.$$Again, relative smallness of $C_{\alpha_{1}k_{0}}$ and $C_{\alpha_{2}k}$ implies relative prohibition of the transition of $|k_{0}\rangle \to |k\rangle$. By definition, one may require that the average shape $\langle |C_{\alpha k^{\prime }}{|}^{2}\rangle$ should change slowly (or remains a constant) within one degenerate subspace. Then, such as in the nondegenerate case discussed above, one still finds that $\langle |C_{\alpha_{1}k_{0}}{|}^{2}\rangle ≃\langle |C_{\alpha_{2}k}{|}^{2}\rangle$ and obtains an equation similar to Equation (Equation 23).

When the unperturbed spectrum has a weak degeneracy such that each set $S_{k}$ contains a small number of elements, the effect induced by the degeneracy is small. Then, according to discussions given above, still, relative smallness of the rescaled components usually implies relative smallness of the transition probability divided by the average shape of EFs. Furthermore, as a result, high population of small rescaled components is related to strong prohibition of quantum transition.

In the rare case that the degeneracy is very high such that the set $S_{k}$ contains a large number of elements, the situation is more complicated in the quantitative aspect. In fact, the perturbation is extremely strong within each degenerate subspace and this may suppress the number of very small components ${\tilde{C}}_{\alpha k}$. Nevertheless, qualitatively, high population of small rescaled components is still related to strong prohibition of quantum transition. (Here, we do not discuss the very special case, in which the spectrum of $H^{0}\left(0\right)$ is completely degenerate. In this case, it is unnecessary for $g(\tilde{C})$ to be extraordinarily large at $\tilde{C}=0$.)

Summarizing the above discussions, we obtain the following result, i.e., the value of g(0) gives a relative measure to the extent of prohibition of quantum transition induced by the perturbation ϵV. Usually, larger value of g(0) may be related to stronger prohibition; while, this expectation may be valid only qualitatively for highly degenerate levels [50]. (As is known, certain elements of the AGP may also be related to transition probability amplitudes. However, the norm of AGP does not show this feature in a direct way.)

### 3.2. g(0) in Systems from Integrable and Chaotic

In this section, we discuss properties of g(0) in chaotic and integrable systems.

First, we discuss $H^{0}\left(\lambda \right)$ as a quantum chaotic system, which satisfies the ETH [13,24,51,52,53,54,55,56]. (see Ref. [57] for a semiclassical proof of ETH.) Technically, the ETH is written as an ansatz for a special structure of the matrix of an observable *O* on the energy basis, i.e.,
(24)$$O_{kk^{\prime }}=f\left(e_{k}\right)\delta_{kk^{\prime }}+g\left(e_{k},e_{k^{\prime }}\right)r_{kk^{\prime }},$$
where f(e) and $g(e,e^{\prime })$ are smooth functions of their variables, $r_{kk^{\prime }}=r_{k^{\prime }k}^{*}$ are independent random variables with normal distribution (zero mean and unit variance), and $g^{-2}$ scales as the density of states $\rho_{\mathrm{dos}}$ with the system size. Under a perturbation *V* that satisfies the ETH ansatz in Equation (Equation 24) and under a very small ϵ such that Equation (Equation 19) is valid, it is easy to compute the average shape of EFs from Equation (Equation 19), obtaining that
(25)$$\langle |C_{\alpha k}{|}^{2}\rangle ≃\frac{\epsilon \;g(e_{k},e_{k_{\alpha }})}{|E_{k_{\alpha }}^{0}-E_{k}^{0}|}.$$Then, noting the randomness of $r_{kk^{\prime }}$, one sees that the distribution of these rescaled components should have a Gaussian form, as illustrated numerically in Ref. [33]. Hence, $g\left(0\right)=1/\sqrt{2\pi }\approx 0.4$ [cf. Equation (Equation 15)] in quantum chaotic systems.

Next, we discuss the integrable case, namely $H^{0}\left(0\right)$. As discussed previously, we consider only those perturbations *V*, whose matrix $\left[V_{kk^{\prime }}\right]$ in the integrable basis {|k(0)〉} has a sparse structure, with many zero elements. To figure out properties of g(0) under a very small ϵ, let us consider an arbitrary perturbed state |α〉. We need to discuss two cases separately, as done below, depending on whether the unperturbed level, which is the closest to $E_{\alpha }$, is degenerate or not:In the case that the level is nondegenerate, Equation (Equation 19) is valid. Then, clearly, the sparse structure of $\left[V_{kk^{\prime }}\right]$ implies that the distribution $g(\tilde{C})$ should have a high peak at $\tilde{C}=0$, with g(0)≫1.In the case of a degenerate level, let us use $|k_{\alpha }^{\eta }\rangle$ (with η=1,2,…) to indicate those states |k〉 that correspond to this unperturbed level. It is straightforward to generalize the perturbative treatment given in the above section and obtain an equation similar to Equation (Equation 19), but, for $k\neq k_{\alpha }^{\eta }\;\left(\forall \eta \right)$. Then, one reaches a similar conclusion that the distribution $g(\tilde{C})$ should have a high peak at $\tilde{C}=0$, usually with g(0)≫1.

It is useful to give further discussions on the computation of the average shape $\langle |C_{\alpha k}{|}^{2}\rangle$ in the degenerate case. There are two methods that may be adopted. The first method is to exclude all the components $C_{\alpha k}$ for the degenerate subspace, i.e., for $k=k_{\alpha }^{\eta }$. This method, though favorable analytically, may face the following subtle problem in numerical simulations, which is induced by the fact that the parameter ϵ always has some lower bound in numerical computations, i.e., there may exist nearly degenerate levels, which are effectively degenerate in numerical simulations with a finite ϵ.

To avoid the above-discussed subtleness in numerical simulations, one may adopt a second method, though it is not so attractive analytically. This method excludes only those components that satisfy $|C_{\alpha k}{|}^{2}>\Lambda_{b}$, where $\Lambda_{b}$ is some (adjusting) parameter that is introduced for obtaining a smooth average shape of EFs. (Meaningful results should be insensitive to the exact value of $\Lambda_{b}$ within a reasonable region. We checked this point in our numerical simulations to be discussed later.) When the degeneracy is low, there is usually only a small difference between the values of g(0) obtained by the two methods. While, the difference may be large when the degeneracy is high, with the former value usually being larger than the latter.

In fact, the above-discussed differences corresponding to different situations with degeneracy is not crucial for our purpose here. What is of practical importance is that, in all the cases, the distributions $g(\tilde{C})$ show high peaks at $\tilde{C}=0$, which is the characteristic feature of integrability. Moreover, we recall that, as discussed in the above section, high degeneracy may increase the difficulty in quantitative evaluation of g(0). For these reasons, in characterizing integrability-chaos crossover, the quantity $g^{-1}\left(0\right)$ may be practically better than g(0). In fact, most of the indefiniteness, if not all of them, are restrained in $g^{-1}\left(0\right)$. The quantity $g^{-1}\left(0\right)$ is expected to satisfy $g^{-1}\left(0\right)≃\sqrt{2\pi }$ for chaotic systems, while, is small for integrable systems. (The exact value of the smallness of $g^{-1}\left(0\right)$ is not so important in distinguishing between integrability and chaos, though it may be informative for other purposes).

Based on the above discussions, loosely speaking, one may expect the following picture for behaviors of the quantity $g^{-1}\left(0\right)$ in an integrability-chaos crossover. At λ=0, one expects that $g^{-1}\left(0\right)\approx 0$, except for the case of very high degeneracy. With λ increasing from 0, the EFs of |k(λ)〉 on the integrable basis {|k(0)〉} become more and more complicated. As a consequence, some previously very small elements $V_{kk^{\prime }}$ become nonnegligible and, then, as predicted by Equation (Equation 19), more and more rescaled components may obtain nonnegligible values. This implies that, with increasing λ, $g^{-1}\left(0\right)$ should increase, until it reaches a saturation value of $\sqrt{2\pi }$, which corresponds to quantum chaos.

## 4. Numerical Simulations

In this section, we present numerical simulations that have been performed in a three-orbital Lipkin-Meshkov-Glick (LMG) model [49,58], illustrating g(0) as a measure for characterizing the integrability-chaos crossover.

### 4.1. The model

The LMG model consists of Ω fermions, which may occupy three single-particle energy states labeled by s=0,1,2. We use $\eta_{s}$ to denote the energy of the *s*-th single-particle state and, for brevity, we set $\eta_{0}=0$. In this paper, we are interested in the collective motion of this model, for which the dimension of the Hilbert space is $\frac{1}{2}\left(\Omega +1\right)\left(\Omega +2\right)$. The classical counterpart of this collective motion has a two-dimensional configuration space.

The Hamiltonian of the model is constructed from the following operators $K_{rs}$,
(26)$$K_{rs}=\sum_{\gamma =1}^{\Omega }a_{r\gamma }^{†}a_{s\gamma },\;r,s=0,1,2,$$
where $a_{r\gamma }^{†}$ and $a_{r\gamma }$ are fermionic creation and annihilation operators obeying the usual anti-commutation relations. The operators $K_{ss}$ in fact represent the particle-number operators for single-particle states *s*, while, $K_{sr}$ with s≠r are level raising/lowering operators. The Hamiltonian is written as
(27)$$\begin{array}{cc} & H^{0}\left(\lambda \right)=\eta_{1}K_{11}+\eta_{2}K_{22}+\lambda V, \end{array}$$
where
(28)$$\begin{array}{cc} & V=\sum_{t=1}^{4}\mu_{t}V^{\left(t\right)}. \end{array}$$In Equation (Equation 28), $\mu_{t}$ are parameters and
(29a)$$V^{\left(1\right)}=K_{10}K_{10}+K_{01}K_{01},$$
(29b)$$V^{\left(2\right)}=K_{20}K_{20}+K_{02}K_{02},$$
(29c)$$V^{\left(3\right)}=K_{21}K_{20}+K_{02}K_{12},$$
(29d)$$V^{\left(4\right)}=K_{12}K_{10}+K_{01}K_{21}.$$

For symmetric states in the collective motion, the operators $K_{rs}$ may be written in terms of bosonic creation and annihilation operators denoted by $b_{s}^{†}$ and $b_{s}$ [59],
(30a)$$K_{rs}=b_{r}^{†}b_{s},$$
(30b)$$K_{r0}=K_{0r}^{†}=b_{r}^{†}\sqrt{\Omega -b_{1}^{†}b_{1}-b_{2}^{†}b_{2}},$$
for r,s=1,2.

The classical counterpart may be obtained by making use of the following transformation,
(31)$$b_{s}^{†}=\sqrt{\frac{\Omega }{2}}\left(q_{s}-ip_{s}\right),\;\;\;b_{s}=\sqrt{\frac{\Omega }{2}}\left(q_{s}+ip_{s}\right).$$

It is easy to verify that $q_{r}$ and $p_{s}$ obey the following commutation relation,
(32)$$\left[q_{r},p_{s}\right]=\frac{i}{\Omega }\delta_{rs}.$$Thus, 1/Ω plays the role of an effective Planck constant, $\hbar_{\mathrm{eff}}=\frac{1}{\Omega }$, and the classical limit of the model is obtained by letting Ω→∞. In the classical Hamiltonian, parameters take the form of $\eta_{s}^{\mathrm{cl}}=\eta_{s}\Omega$ and $\mu_{t}^{\mathrm{cl}}=\mu_{t}\Omega^{2}$. Quantum systems corresponding to the same values of $\eta_{s}^{\mathrm{cl}}$ and $\mu_{t}^{\mathrm{cl}}$ share a common classical counterpart.

### 4.2. Numerical Results

As is well known, on the integrable side, a model may behave differently under different winding numbers, particularly, at rational and irrational values. In the LMG model, the winding number is given by the ratio of $\eta_{1}/\eta_{2}$. in our numerical simulations, we have studied the crossovers starting from several integrable Hamiltonians, corresponding to different values of $\eta_{1}$ and $\eta_{2}$. We have the freedom of keeping $\eta_{1}$ and $\eta_{2}$ in the order of magnitude of 1. Specifically, we have studied two rational values of $\eta_{1}/\eta_{2}$ and two irrational ones: $\left(\eta_{1},\eta_{2}\right)=\left(1,2\right),\left(1,3\right),\left(\sqrt{3},2\right)$ and $\left(\sqrt{5}-1,2\right)$.

On the chaotic side, parameters in *V* are chosen such that this observable has essentially the same classical limit as that studied in Ref. [30]; specifically, $\mu_{1}=0.0077625,\;\mu_{2}=0.008775,\;\mu_{3}=0.0095625$, and $\mu_{4}=0.0082125$. Other parameters used in our simulations are $\epsilon =10^{-4}$ for small perturbation and the total particle number Ω=200, for which the dimension of the Hilbert space is 20301. Under these parameters, the mean value of $|\epsilon V_{kk^{\prime }}|$ of nonzero elements of *V* in the integrable basis is about $2.6\times 10^{-3}$, obviously smaller than the mean level spacing which is about $10^{-2}$, satisfying the requirement of weak perturbation.

In the computation of the average shape $\langle |C_{\alpha k}{|}^{2}\rangle$ (Equation (Equation 6)), we adopted the second method discussed previously, with $\Lambda_{b}=0.1$; i.e., all big components with $|C_{\alpha k}{|}^{2}\geq 0.1$ were excluded. The width δe was adjusted such that each window $\Gamma_{\alpha }$ includes 16 levels. In the computation of g(0), for each system $H^{0}\left(\lambda \right)$, 2000 perturbed states |α〉 lying in the middle energy region were used, and for each |α〉, 2000 components ${\tilde{C}}_{\alpha k}$ with $E_{k}^{0}$ around $E_{\alpha }$ were used.

Numerical results are shown in Figure 1, Figure 2, Figure 3 and Figure 4 for the four pairs of $\left(\eta_{1},\eta_{2}\right)$ mentioned above, respectively. It is seen that the $g^{-1}\left(0\right)$-plots in all the four figures show a same pattern of behavior from integrability to chaos. Furthermore, this pattern is much simpler than those of g(0), as explained previously due to the fact that the difference among big values of g(0) is restrained in $g^{-1}\left(0\right)$. For the sake of comparison with standard spectral-statistics analysis, we also plot the quantity $\Delta_{W}$ in Equation (Equation 16) as a measure for the distance to chaos. Furthermore, moreover, we plot the following quantity as a measure for the distance to integrability, i.e.,
(33)$$\Delta_{P}=\int \left|I\left(s\right)-e^{-s}\right|ds,$$
where $e^{-s}$ is the Poison distribution expected for the nearest-level-spacing distribution in generic integrable systems.

As is known, the parameter region (0,1) for a integrability-chaos crossover may usually be divided into three subregions: a nearly integrable regime, a nearly chaotic regime, and a crossover regime. A merit of the quantity $g^{-1}\left(0\right)$ is that it shows this division in a very clear way, the details of which we discuss below.

A nearly integrable regime, in which the value of $g^{-1}\left(0\right)$ is close to 0. In this regime, the perturbation-induced transition is strongly prohibited between many of the unperturbed states.A nearly chaotic regime, in which $g^{-1}\left(0\right)\approx \sqrt{2\pi }$. In this regime, the perturbation-induced transition is not prohibited in a statistical sense.An intermediate (cross-over) regime, in which $g^{-1}\left(0\right)$ increases “rapidly”, approximately from 0 to 2.5.

Specifically, in the case of $\left(\eta_{1},\eta_{2}\right)=\left(1,2\right)$, as seen in the upper panel of Figure 1, the above-discussed three regimes occupy approximately the three subregions of (0,0.22),(0.5,1) and (0.22,0.5), respectively. In this figure, the point of λ=0 looks quite special, because the value of $g^{-1}\left(0\right)$ at λ=0 is not small, but, about 0.5. (One should note that, although the value of g(0) at λ=0 in this figure is not very large, about 2, it is still obviously larger than the value expected for chaotic systems, the latter of which is about 0.4.) This phenomenon is due to the high degeneracy of the spectrum of the integrable system $H^{0}\left(0\right)$ with $\left(\eta_{1},\eta_{2}\right)=\left(1,2\right)$. In fact, with $\Lambda_{b}=0.1$, most of the components $C_{\alpha k}$ of |k〉 lying in highly degenerated subspaces were included in the computation of g(0) and, since the perturbation is effectively extremely strong within each degenerate subspace, the number of very small components in them is not large. Please note that this phenomenon disappears with a little increase of λ, which destroys the degeneracy, and as a result the value of $g^{-1}\left(0\right)$ becomes quite small, as seen in the figure.

In Figure 2, with $\left(\eta_{1},\eta_{2}\right)=\left(1,3\right)$, the three regimes are seen clearly, too, approximately occupying (0,0.46), (0.74,1), and (0.46,0.74), respectively. In this figure, the value of $g^{-1}\left(0\right)$ at λ=0 is not very small, either, about 0.18. This value is smaller than that in Figure 1, because the degeneracy in the system with $\left(\eta_{1},\eta_{2}\right)=\left(1,3\right)$ is not as high as that in the case of $\left(\eta_{1},\eta_{2}\right)=\left(1,2\right)$. Consistently, in Figure 3 and Figure 4, the values of $g^{-1}\left(0\right)$ are already small at λ=0 for integrable systems $H^{0}\left(0\right)$ possessing nondegenerate spectra.

In the lower panels of the four figures, it is seen that the g(0)-plots show rich behaviors in the nearly integrable regime. In the case with rational ratios $\eta_{1}/\eta_{2}$ (Figure 1 and Figure 2), g(0) is relatively small at λ=0 (corresponding to the above-discussed relatively large values of $g^{-1}\left(0\right)$) and increases immediately with a little increase of λ. For irrational ratios $\eta_{1}/\eta_{2}$ (Figure 3 and Figure 4), although g(0) are already not small at λ=0, they also undergo immediate rapid increase with a little increase of λ.

Correspondingly, from the g(0)-plots, one also sees three subregions in each figure — nearly integrable, intermediate (crossover), and nearly chaotic. In Figure 1, Figure 2, and Figure 4, the intermediate subregions are close to those obtained from the level-statistics properties ($\Delta_{W}$). However, notable differences between them are seen in Figure 3, showing the information supplied by g(0) for the crossover is not always in quantitative agreement with that obtained from the level statistics. We are not able to provide further discussion on such a difference, nor on rich behaviors of g(0) on the integrable side, because as discussed previously, further analytical study is needed for a deeper understanding of them.

## 5. Conclusions

In this paper, a measure has been proposed for characterizing integrability-chaos crossover for a big class of quantum models of practical interest, which possess underlying dynamic Lie groups. To compute this measure, a very small perturbation (ϵV) of physical relevance is applied to the studied system and components of perturbed states on the unperturbed basis are rescaled with respect to the average shape of eigenfunctions. The measure is given by g(0), the value of the distribution of rescaled components at the origin point, or, practically by $g^{-1}\left(0\right)$. In a relative sense, g(0) has the physical meaning of describing the extent of prohibition of ϵV-induced transitions between unperturbed states.

As is known, the whole parameter region of integrability-chaos crossover may usually be divided into three subregions: a nearly integrable regime, a nearly chaotic regime, and a crossover regime. Different measures employed usually give divisions with some differences. Numerical simulations performed in the LMG model, which possesses a classical counterpart in a two-dimensional configuration space, show that the behavior of 1/g(0) gives such a division in quite a clear way. The division is in qualitative consistency with that obtained from spectral statistics, but not always quantitatively. Future investigations are needed for analytical understanding of the difference.

In principle, the proposed measure may be used in both cases with a noninteracting integrable system and with an interacting integrable system. In this paper, numerical simulations are given only in a simple model of the former type. In future, it would be of interest to see its application to models of the latter type, which might be more complicated than the former type. Moreover, application of the proposed measure to many-body systems, particularly on the nearly integrable side, may be an interesting topic for future study, too.

## Figures and Tables

**Figure 1 entropy-25-00366-f001:**
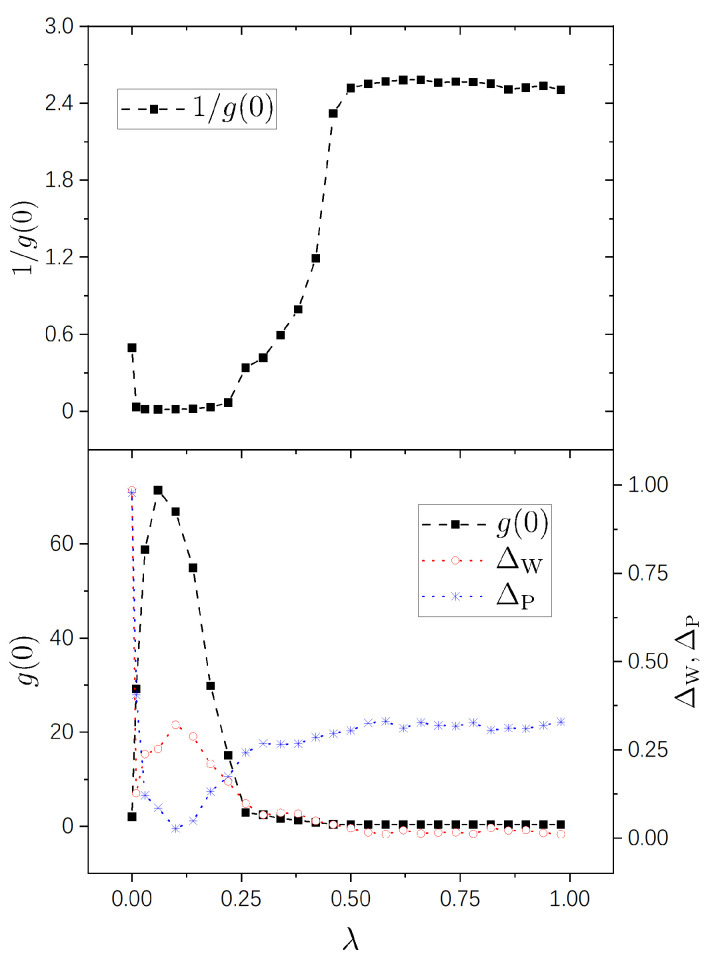
Variation of $g^{-1}\left(0\right)$ (upper panel), and g(0) (lower panel) versus λ (solid squares connected by dashed lines) for $\eta_{1}=1,\eta_{2}=2$. Other parameters: $\mu_{1}=0.0077625,\;\mu_{2}=0.008775,\;\mu_{3}=0.0095625$, $\mu_{4}=0.0082125,\Omega =200$, and $\epsilon =10^{-4}$. For comparison, variation of $\Delta_{W}$ in Equation (Equation 16) (empty circles connected by dotted line (red)) and of $\Delta_{P}$ in Equation (Equation 33) (stars connected by dotted line (blue)) are also plotted.

**Figure 2 entropy-25-00366-f002:**
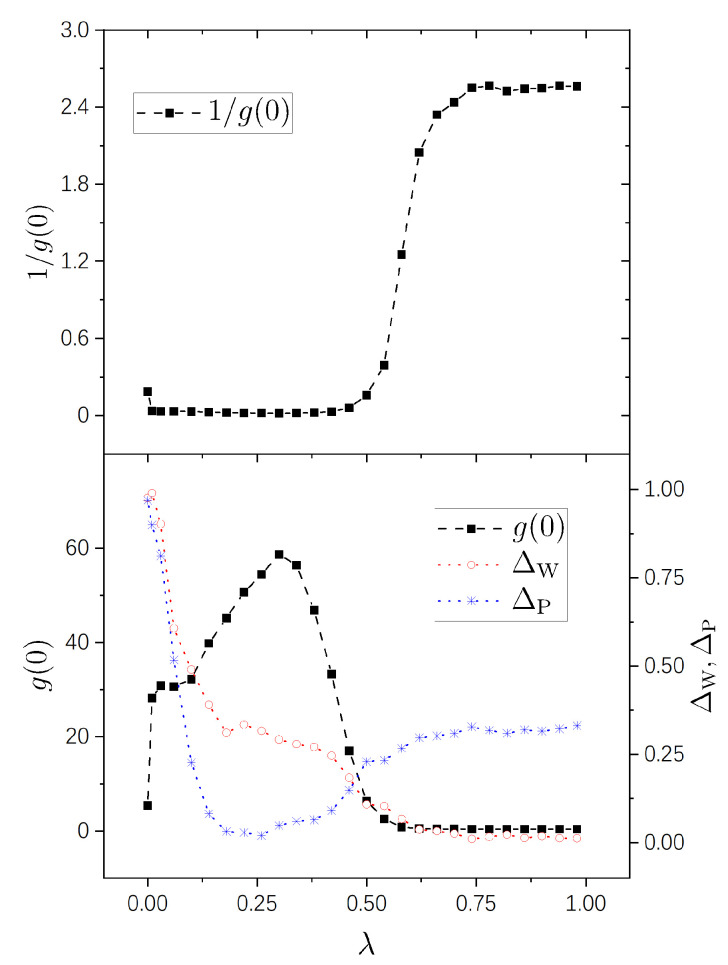
Similar to Figure 1, but for $\eta_{1}=1$ and $\eta_{2}=3$.

**Figure 3 entropy-25-00366-f003:**
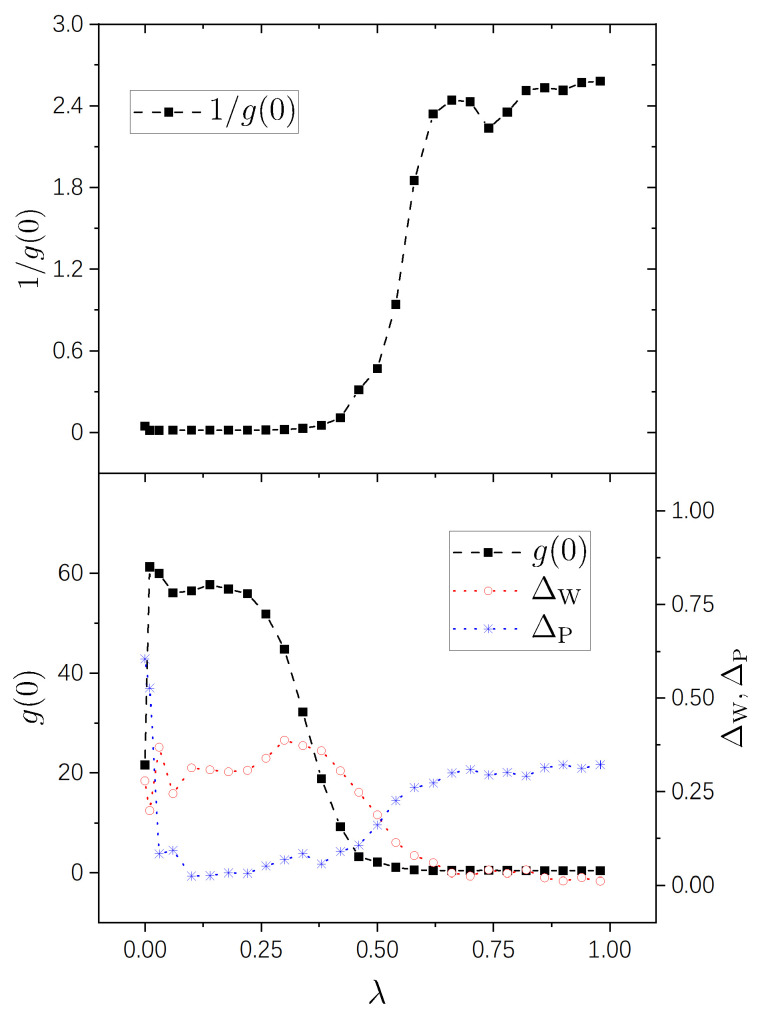
Similar to Figure 1, but for $\eta_{1}=\sqrt{3}$ and $\eta_{2}=2$.

**Figure 4 entropy-25-00366-f004:**
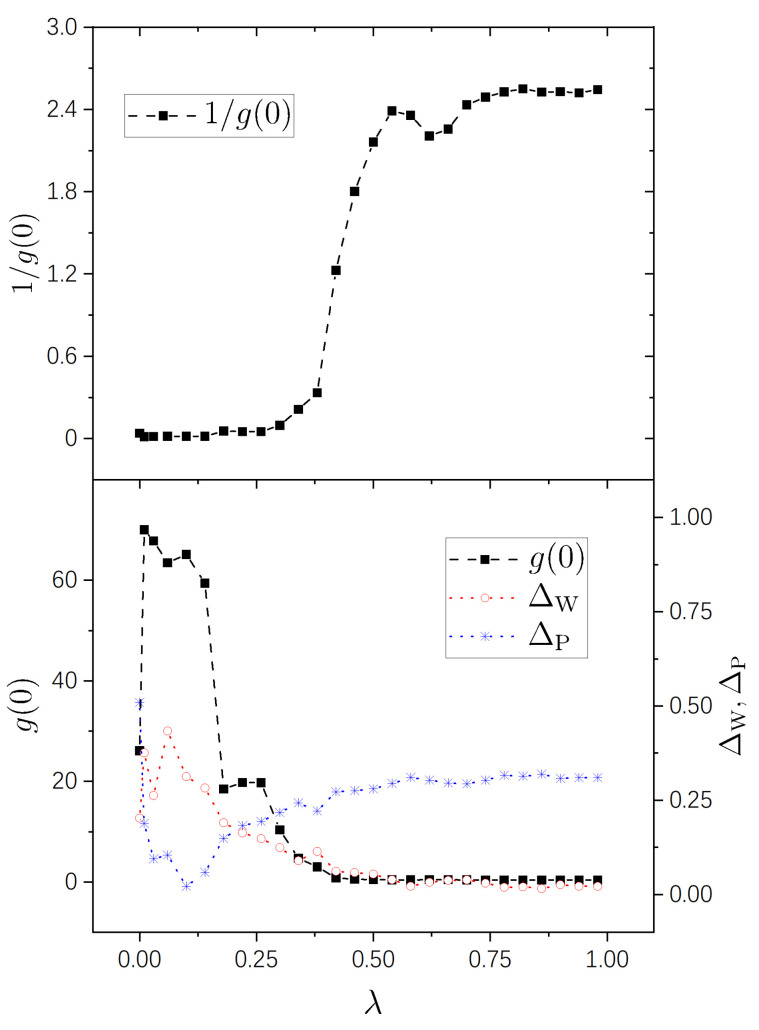
Similar to Figure 1, but for $\eta_{1}=\sqrt{5}-1$ and $\eta_{2}=2$.

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
