# Peer review of "A Physical Measure for Characterizing Crossover from Integrable to Chaotic Quantum Systems"

_entropy, 2023, doi:10.3390/e25020366_

Round 1

Reviewer 1 Report

In this work, the authors consider a new measure for characterizing the crossover from chaotic to integrable systems, based on the distribution of matrix elements of eigenstates in an unperturbed basis. The authors motivate this measure by comparison with previous diagnostics of quantum chaos (the adiabatic gauge potential/eigenstate susceptibility approach and the deviation from a Gaussian distribution of these matrix elements). This measure is argued to be more 'physical' by relating it to the probability of quantum transitions induced by a perturbation. This approach is interesting, but I have some major comments before I can recommend this work for publication.

- Throughout the manuscript the authors argue that integrable models should lead to matrix elements of a perturbation V that have a sparse structure. However, this very much depends on 1) the choice of perturbation, and 2) the integrable model being non-interacting. It is always possible to find a perturbation that does not have a sparse structure, and for interacting integrable models perturbations will generally not have a sparse structure (as also important in Ref. [20], where a distinction is made between interacting and nointeracting integrable). For the measure to be a valid indicator of quantum chaos versus integrability, the authors should consider interacting integrable models such as the Heisenberg XXZ model and argue the importance of the choice of perturbation V.

- The adiabatic gauge potential approach is similarly connected to the probability of quantum transitions, as detailed in e.g. M. Kolodrubetz, D. Sels, P. Mehta, A. Polkovnikov, Geometry and non-adiabatic response in quantum and classical systems, Physics Reports 697 1–87 (2017). Since the authors argue that the proposed measure is more physical because of such a connection, it would be useful to also comment on this connection and argue the distinction.

- The authors emphasize that their measure can be used to characterize the 'crossover' from integrability to chaos. However, such a crossover is typically only expected in few-body systems, since integrability of many-body Hamiltonians is immediately broken upon the introduction of a small perturbation and the notion of 'chaos' then depends on the specific measure used. Can the authors comment on what would happen in the thermodynamic limit or how their proposed measure depends on the system size?

- A bibliographic note: The authors might want to also consider M. Brenes, T. LeBlond, J. Goold, and M. Rigol, Eigenstate Thermalization in a Locally Perturbed Integrable System, Phys. Rev. Lett. 125, 070605 (2020) and relevant references therein. There are also a series of works by Lea Santos and various co-authors discussing different measures of chaos such as the 'correlation hole' that the authors might want to consider.

Author Response

Thanks for your comments.

Please see the attachment for our responses.

Reviewer 2 Report

In this paper the Authors study the transition from integrability to chaos in many-body quantum systems. After a short summary of the existing methods utilised in literature to characterise this transition they propose a 

new one which consists in the study of the response of eigenstates to a small generic perturbation.

They claim this method characterised by the use of an intrinsic basis thus avoiding the problem of basis-dependence 

of other methods where statistical properties of eigenfunction have been considered, e.g. Refs 12-19.

Moreover, contrary to the standard approach involving spectral properties, which are basis independent, they claim their own approach to be more physical.

Let me first comment of these two issues. First of all, it is obviously true that while spectral properties are basis independent, properties of eigenfunctions depends on the basis chosen. This is a very long debated issue since the birth of quantum chaos. Can I say : so what? A lot of physics is basis-dependent, think about Anderson localisation of Gibbs, Fermi-Dirac of Bose-Einstein  distribution and apparently nobody is worried about that. So, please let us stop to state that this is a problem.

Second point, more serious : Why do we need a criterium for the regular-to-chaos transition? In my opinion this should be a crucial point in any analysis concerning quantum many-body system. The Authors didn't discuss this important issue. I strongly suggest them to include a discussion about this point. Of course this is quite a matter of taste, it could be the foundations of statistical mechanics, or the problem of thermalisation, or the definition of ergodicity, etc etc.

Third point : Of course establishing a threshold (if any)  could be very useful to see, for instance, when a statistical approach can be used for instance to predict relaxation times or ensemble averages. But the method should be easy, simple and possibly experimentally accessible. I might agree with the Authors that levels statistics even if simple, and relatively easy (if we avoid the issues related to the spectrum unfolding) is surely not so easy to implement from the experimental point of view. And the same critic can be applied to all those approaches involving the properties of eigenfunctions. Considering their approach I don't see any improvement with respect to the  present state-of-art, from this perspective.

To be honest it seems to me even more complicated and rough, especially when arbitrary cut-off ($\Lambda_b$) are introduced as adjusting parameters. Even the discussion about the spectrum degeneracy makes the whole story more confusing : What does it mean that g(0) has not a clear physical meaning in presence of degeneracy?

Last but not least they applied their theory only to the three orbital LMG model and, more important, they compare their results with the $\Delta_W$ parameter coming from standard spectral statistics only.  I don't see any appreciable difference between the two parameters. As usual we may identify a regular region, a chaotic one and a smooth transition between the two guys. So what? What are the physical features characterising the intermediate region? No question, no answer.

I also have two other small issues concerning their paper.

1) in page 1 line 19-20 they claim that "percentage of phase space, which is occupied by chaotic trajectories, supplies a good quantitative measure in the study of crossover from integrability to chaos".

I think that they do have in mind two-dimensional classical systems here. The situation in general many-body classical system is far from being trivial (and solved). I suggest the Author to comment on this point.

2)page 7 lines 158-159 they write "     We recall that the matrix [Vkk′ ] in  the integrable basis {|k(0)⟩} has a sparse structure, with many zero elements". 

It is not clear to me why the matrix elements of the same perturbation in the chaotic basis {|k(1)} should be different.

Could they explain better this point maybe showing by specific numerical examples?

In conclusion I think the paper, despite all my critics,  is still worth of publication, provided the Authors discuss the points raised in my report above.

Author Response

(The authors gave the same response as above.)

Round 2

Reviewer 1 Report

I appreciate the authors' efforts in revising the manuscript. The revised version has addressed most of my concerns, but I have some remaining issues with the added Section. These comments serve to further clarify the distinction between interacting and non-interacting integrable systems, and I can recommend the paper for acceptance provided these are addressed.

As is known, if no restriction is imposed to the operator V, generically, no qualitative difference may be guaranteed between properties of the components C_αk in integrable systems and those in chaotic systems.

I don’t believe this to be true, see e.g. the results from Phys. Rev. B 102, 075127 (2020) by Brenes et al., where the matrix elements of perturbations in integrable and chaotic models are compared. Indeed, if the components of the operator V in the eigenbasis of an integrable Hamiltonian would be the qualitatively similar to its components in the eigenbasis of a chaotic model, integrable models would satisfy the eigenstate thermalization hypothesis – since this is exactly a statement about matrix elements of observables/perturbations in the eigenbasis of the Hamiltonian.

Furthermore, the phrasing of “on a whatever basis” should probably be changed to “in a random basis”.

To give an analysis to the above-discussed point, we note that most physical models of realistic interest have some underlying dynamic Lie groups.

It would be good to point out that this only holds for noninteracting integrable systems, which does not seem to correspond tot "most physical models of realistic interest", unless I’m missing something.

In the paragraph on interacting integrable systems the authors write “While, for a perturbation V, which may bring a limited change to good quantum numbers of the states |k(0)⟩ (usually not the integers mη ), one still finds a sparse structure of the matrix of ⟨k′(0)|V|k(0)⟩.”

Could the authors give an example of such a perturbation? I believe such operators would require the system to be noninteracting. As written, this paragraph is somewhat confusing – is it possible to find such operators for interacting integrable systems or not?

Finally, if the proposed probe can not be used to distinguish interacting integrable from chaotic models, this should also be made clear in the abstract.

Author Response

Thank you very much for your useful suggestions.

We have revised the manuscript based on your comments.

The details of changes and our reply to your questions are in the word file.

Reviewer 2 Report

The Authors have significantly improved the paper and answered to my questions. The paper can be published in the present form.

Author Response

Thank you for reviewing our manuscript and giving useful suggestions. 

Round 3

Reviewer 1 Report

I am satisfied with the author's response and can recommend this paper for publication.